# Factors Associated with Medication Adherence among Patients with Type 2 Diabetes Mellitus: A Hospital-Based Cross-Sectional Study in Nepal

**DOI:** 10.3390/ijerph20021537

**Published:** 2023-01-14

**Authors:** Pushpanjali Shakya, Archana Shrestha, Biraj Man Karmacharya, Donald E. Morisky, Bård Eirik Kulseng

**Affiliations:** 1Department of Public Health and Nursing, Faculty of Medicine and Health Sciences, Norwegian University of Science and Technology, 8905, N-7491 Trondheim, Norway; 2Department of Public Health and Community Programs, Kathmandu University School of Medical Sciences, Dhulikhel 45200, Nepal; 3Institute for Implementation Science and Health, Kathmandu 44600, Nepal; 4Department of Chronic Disease Epidemiology, Yale School of Public Health, New Haven, CT 06520-0834, USA; 5Department of Community Health Sciences, UCLA Fielding School of Public Health, Los Angeles, CA 90095-1772, USA; 6Centre for Obesity Research, Department of Surgery, St. Olavs Hospital—Trondheim University Hospital, N-7006 Trondheim, Norway; 7Department of Clinical and Molecular Medicine, Faculty of Medicine, Norwegian University of Science and Technology, N-7491 Trondheim, Norway

**Keywords:** adherence, factors, Nepal, medication, type 2 diabetes

## Abstract

As diabetes increases globally, high mortality increases due to complications of uncontrolled sugar. Medication adherence is important to control blood sugar and prevent its complications. Objective of the study was to identify factors associated with medication adherence among type 2 diabetes patients. A cross-sectional study was conducted among 343 patients visiting Dhulikhel Hospital, Nepal, for their fasting blood sugar test from September to December 2016. Inclusion criteria: patients with type 2 diabetes, under diabetes medication for past three months (minimum), age ≥ 18 years. The outcome of the study was medication adherence measured using the eight-item Morisky medication adherence scale (MMAS-8) (© 2006 Donald E. Morisky). Multivariate logistic regression was used for the analysis. Results showed that 61% of respondents had high medication adherence; adherence was positively associated with formal education [AOR: 2.43 (95% CI: 1.34, 4.39)] and attendance at diabetes counseling [AOR: 1.76 (95% CI: 1.02, 3.04)] after adjusting for age, occupation, medicine intake duration and diabetes medicine types. The study concluded that formal education and attendance at diabetes counseling positively affected patients’ adherence to medicine. We encourage healthcare institutions to provide counseling services to all the patients with type 2 diabetes and focus more on those who are less educated.

## 1. Introduction

Diabetes is a chronic medical condition characterized by high blood sugar level caused by impaired glucose metabolism [1]. The prevalence of diabetes is rapidly increasing worldwide, affecting 537 million adults in 2021 [2]. Diabetes is more prevalent in low-and middle-income countries (LMIC), where over three in four adults are living with diabetes [2]. The second largest prevalence of diabetes was found in South East Asia [2]. The prevalence of diabetes in Nepal was 10% as per systematic review and meta-analysis [3] which was higher than reported in a study conducted in 2020 [4]. Hyperglycemia (high blood glucose) is the third highest risk factor for global premature mortality [5] causing 6.7 million deaths in 2021 [2]. There has been a 3% increase in age-standardized diabetes mortality rates from 2000 to 2019 globally [1]. Further, the increase in mortality rate due to diabetes has increased by 13% in LMIC [1]. The all-age diabetes mortality rate has doubled between 1990 to 2019 in Nepal [6]. The mortality is caused by life-threatening complications associated with diabetes such as kidney failure, heart attacks, and stroke [1]. Disability-adjusted life years (DALYs) due to diabetes have also increased from 1990 to 2019 worldwide [7] and also in Nepal [6].

Diabetes has imposed a significant economic burden in the world with an expenditure of 966 billion dollars [2]. The economic burden has been encountered by the individuals, their families, subsequently countries and their national health care systems [2]. The cost of diabetes is more challenging for LMIC such as Nepal, where a single patient with diabetes paid around 13.30 US Dollars (USD) per out-patient clinic visit while the total cost for diabetes treatment and care per month costs around USD 40.40 [8]. Most of the Nepalese population cover this cost out-of-pocket [9], although some medicines are offered free of cost at government health care centers. The cost of diabetes medications has been considered to be one of the barriers to achieving targeted glycemic control [10]. On the other hand, sustained glucose control is important to minimize the cost of diabetes care and prevent its complication [11]; it can be achieved by good medication adherence along with adhering to proper diet control, recommended exercise and lifestyle modification. However, increased cost of medication such as cost-related insulin underuse has resulted in the non-adherence of medication among patients with diabetes [10,12]. Consequently, poor medication adherence leads to poor glycemic control [13,14] and, over time, uncontrolled blood sugar leads to diabetes related micro or macro complications [15,16].

In a global context, diabetes medication adherence has been associated with the female gender [17], level of education [18,19], knowledge about diabetes [18,19], duration of diabetes [19,20], perception of disease severity [17], among other factors. In Nepal, few studies have identified factors for the adherence or non-adherence to treatment among patients with type 2 diabetes [21,22,23,24,25]. The good medication adherence measured using the eight-item Morisky medication adherence scale (MMAS-8) was 28.5% in Nepal [21]. The identified associated factors of non-adherence to diabetic treatment found in Nepelese studies were: age [23], gender [24], educational status [23,24], occupation [24], monthly income [23], working group [23], duration of diabetes [23], while diabetic counseling was associated with good medication adherence [21]. The reasons for non-adherence were intentional discontinuation, forgetfulness, carelessness, and hypoglycemia [22]. However, these previous studies have used simple statistical methods to determine predictors of medication adherence [21,22,24,25]; only one used multiple logistic regression [23]. Two studies used MMAS-8 [21,25], two studies used MMAS-4 [23,24], one study did not use standardized questions to measure medication adherence and failed to identify its associated factors scientifically [22]. One qualitative study was also conducted investigating factors impacting medication in a Nepalese population with type 2 diabetes [26].

It is important to measure medication adherence and its associated factors among patients with type 2 diabetes in Nepal in a valid and reliable way, using standardized tools and using advanced statistical analysis. Hence, the eight-item Morisky medication adherence scale (MMAS-8) (© 2006 Donald E. Morisky) [13,27] was used in this study to measure medication adherence among patients with type 2 diabetes in an urban tertiary level hospital in Nepal. Further, the present study used multivariate logistic regression to quantify the association between factors with medication adherence by controlling the effect of confounders. Since studies of such types in the context of Nepal have been limited, the present study aimed to identify factors associated with diabetes medication adherence among patients with type 2 diabetes in Nepal. The findings of this study will help to gain insights into the predictors of medication adherence among patients with type 2 diabetes in Nepal. Additionally, the findings will be helpful for stakeholders including medical practitioners and policy makers to initiate evidence-based planning for proper management of type 2 diabetes in Nepal.

## 2. Materials and Methods

Study design and study site: We conducted a hospital-based cross-sectional study at the laboratory department of Dhulikhel Hospital among patients with type 2 diabetes visiting for their regular fasting blood sugar (FBS) test, from September to December 2016. Dhulikhel Hospital is a tertiary care hospital located at the center of Kavrepalanchowk district. It provides services to a population of more than 1.9 million in its catchment area. The hospital has a good facility for patients with diabetes, including nurse-run diabetes counseling. The service of diabetes counseling was started in 2012.

Study population and sample size: We recruited a total of 343 eligible patients in this study based on the sample size calculated for a bigger study with glycemic control as an outcome and medication adherence as an exposure [28]. The eligibility criteria for the patients were: (i) had type 2 diabetes, (ii) under diabetes medication for the past three months, (iii) 18 years or older. The participants were operationalized as having type 2 diabetes based on the participants’ self-history of type 2 diabetes; a history of taking diabetes medication and reconfirmation of the diagnosis & medication from their out-patient department registration card. We excluded those with type 1 diabetes, gestational diabetes, and those who were critically ill. Participation was voluntary. 

Participant recruitment: We informed the potential participants about the study using a multi-coloured flex with description of study in Nepali language, posted at the laboratory department door. In addition, we verbally announced about the study at the poster site to draw the attention of potential participants. When the participants came in contact with us, we explained the purpose of the study and received written informed consent from those who agreed to participate. The convenience sampling technique among the participants who visited in the laboratory department of the hospital may have led to selection bias, which needs to be taken into account for interpretation of the results.

Data collection methods and tools: We used face-to-face interview method in the Nepali language to collect the data. We used a pre-tested standardized questionnaire using the Open Data Kit (ODK) software (v1.4.11). The questionnaire includes socio-demographic characteristics, clinical characteristics, eight-item Morisky medication adherence scale (MMAS-8) (© 2006 Donald E. Morisky) [29,30,31,32]. 

### 2.1. Outcome

Our primary outcome of the study was information about medication adherence. The medication adherence was measured using the Nepali version of MMAS-8 (© 2006 Donald E. Morisky) (The MMAS (8-item) content, name, and trademarks are protected by US copyright and trademark laws. Permission for use of the scale and its coding is required. A license agreement is available from Donald E. Morisky, ScD, ScM, MSPH, MMAS Research LLC., donald.morisky@moriskyscale.com).The MMAS-8 score was calculated based on the Morisky guidelines obtained through a license contract. The adherence was categorized as low (total score less than 6); moderate (total score 6 to less than 8); and high (total score equal to 8). 

### 2.2. Exposures

Socio-demographic characteristics, which included age (years), gender (male/female), education (non-formal/up to School Leaving Certificate (SLC)/above SLC), ethnicity (Brahmin/Newars/others), residency (rural/urban), occupation (housewife/business/agriculture/office (professional)/unemployed/other occupation), and family support for medication (yes/no). Clinical characteristics, which included duration of diabetes (years), duration of medication intake (years), immediate family members with diabetes (yes/no), history of hypertension (yes/no), history of current antihypertensive medication (yes/no), history of diabetes complication (yes/no), time given by doctors (minutes per visit), diabetes counseling (yes/no) and types of diabetes medication (Oral Hypoglycemic Agents (OHA)/insulin/both). 

Statistical analysis: We summarized the sample characteristics using mean and standard deviation (SD) of normally distributed continuous variables; median and interquartile range for continuous variables with skewed distribution; and frequency and percentage of categorical variables. We presented all data excluding missing data which resulted in varied sample size for some of the variables like residency and financial support for diabetes medicines. We used multivariate logistic regression to identify factors associated with medication adherence (dependent variable). Only two categories of “low/moderate adherence” (MMAS-8 < 8) = 0 and “high adherence (MMAS-8 = 8) = 1” were used in multivariate logistic regression because there were very few participants in low adherence category.

We used three models for statistical analysis. Model 1 was bivariate analysis. Model 2 was adjusted for sociodemographic variables [age (continuous), education (no formal education = 0/formal education = 1), and occupation (unemployed = 0/employed = 1)]. Model 3 was adjusted for both sociodemographic variables (aforementioned variables) and clinical characteristics [medicine intake duration (natural log), attendance of diabetes counseling (no = 0/yes = 1) and types of diabetes medicines (only OHA = 0/insulin or insulin with OHA = 1)].

We obtained ethical approval from the Regional Committee for Medical and Health Research Ethics (REK) [Ref no. 2016/826/REK midt], Norway; Nepal Health Research Council (NHRC), Nepal [Reg no. 124/2016]; and Institutional Review Committee, Kathmandu University School of Medical Sciences (IRC/ KUSMS), Nepal [Ref no. 96/16]. We also obtained written permission from the hospital director to use the laboratory department of Dhulikhel Hospital. Written informed consent was obtained from all the recruited participants. 

We followed the STROBE (strengthening the reporting of observational studies in epidemiology) checklist [33] to report this cross-sectional study.

## 3. Results

We approached participants until the required sample size of 343 was recruited. A total of 357 were approached, with a response rate of 96%.

Table 1 shows sociodemographic characteristics of the study population. Mean age was 56 years and 54% were male.. Almost half of the participants (49%) were from Newar ethnicity, 90% were currently married, around 63% had formal education, housewife was the most popular occupation (29%) and majority were from an urban area (89%). 

Table 2 presents clinical characteristics stratified by gender. Majority were under only OHA (84.3%). The median medication duration was 3 (IQR: 1, 6) years. The mean Morisky medication adherence scale score (© 2006 Donald E. Morisky) was 7.4 (SD: 1). The results showed that more male participants (76.3%) paid for their medicine themselves compared to the female participants. Around 15% participants had family members with diabetes. Only 42% had received any diabetes counseling. The mean duration for doctor–patient interaction was 8.6 (SD: 4.9) minutes. Nearly half of the participants (47.5%) reported having hypertension and 8.5% reported having chronic complication caused by diabetes.

Figure 1 shows the distribution of level of adherence by gender. The proportion of the total participants with high, moderate and low adherence were 60.9%, 31.8% and 7.3 % respectively. The proportions of medication adherence were similar in males and females. Compared to males, a lower percentage of female patients had high medication adherence. Moderate medication adherence was higher among females than in males and low medication adherence was comparatively higher among males than in females.

Table 3 presents factors associated with medication adherence among patients with type 2 diabetes. High medication adherence was positively associated with having formal education after adjusting for sociodemographic characteristics (age, occupation) and clinical characteristics (medicine intake duration, attendance in diabetes counseling, diabetes medicine types) [OR: 2.4, CI 95%: 1.3–4.4; *p*-value: 0.003]. The odds of high medication were 76% higher among those who attended diabetes counseling sessions compared to those who did not, after adjusting for sociodemographic characteristics and clinical characteristics [OR: 1.7, CI 95%: 1.0–3.0; *p*-value: 0.04]. There was no significant association between medication adherence and age, occupation, medicine intake duration, or diabetes medicine types.

## 4. Discussion

The present study demonstrated that more than half of the study population of hospital-based participants of an urban area in Nepal with type 2 diabetes had high medication adherence. The multivariate logistic regression showed a significant association of medication adherence with level of education and attendance in diabetes counseling. The study did not show association between medication adherence and age, occupation, medication intake duration or types of diabetes medication.

The proportion of high medication adherence in the present study supports the findings in a previous study conducted in the same hospital setting in Nepal which showed 62% for high medication adherence [22]. Similar results were demonstrated in one of the Indian studies [34] and in a study in China [27]. The high adherence in these studies might be because of the hospital-based participants who were more active, more health conscious and hence were more motivated to adhere to their medication. However, the results were in contrast to other Nepalese studies which showed a low proportion of good adherence and a high proportion of non or poor/fair adherence [21,23,24,25]. Another study from an Ethiopian hospital reported a high adherence rate of 48% [19]. The variations in the results might have been attributed to difference of data collection tools, difference in sample size or the true difference in different population. 

In the present study, the mean MMAS-8 (© 2006 Donald E. Morisky) score was 7.4. In a study conducted in China using MMAS-8 for diabetes medicines, the mean medication adherence score was 6.79 (SD: 1.37) [27]. It is important to note that the study in China measured only OHA adherence while the present study measured adherence to both OHA and insulin. Our analysis by type of medicine showed higher adherence among those taking insulin only or insulin with OHA than among those who were under only OHA. It could be due to more caution taken by patients with insulin administration than patients taking only OHA.

In the present study, medication adherence was positively associated with formal education. This was in line with the studies in Nepal which showed lower education led to more non-adherence totherapies including medication [23,25]. The result was also similar to the results of an Ethiopian study which showed that adherence was higher for those with higher educational status than for those with less than grade one [19]. Similar results were also obtained in the studies in India [34] and in Canada [18]. This indicates that better education leads to better medication adherence. Educated individuals could better understand information about the disease and also about the consequences of not adhering to medication. In contrast, the previous study in Nepal did not show any significant association between medication adherence and education [22]. That study was limited by its use of an unstandardized question tool for medication adherence to identify associated factors of medication adherence.

There was a positive association between medication adherence and diabetes counseling in the present study, which was similar to the study conducted in Nepal [21]. It suggests that diabetes counseling helps patients with type 2 diabetes to understand diabetes and highlights the importance of proper diet, regular exercise, regular check-ups and regular medicines. The importance of diabetes management education and support (DSME/S) was also highlighted by the American Diabetes Association, the American Association of Diabetes Educators, and the Academy of Nutrition and Dietetics [35], while Cornell et al. emphasized medication adherence as an integral part of proper diabetes management [36].

Although the present study did not show any significant association between diabetes medication adherence and age, occupation, medicine intake duration or types of diabetes medicines, other studies have demonstrated an association between medication adherence with these factors [23,25,34,37]. Increasing age (≥50 years) has been shown to be associated with lower adherence to medication due to several reasons, for example, the presence of co-morbidities and functional disabilities [23,34,37]. A Nepalese study showed that self-employed participants were more likely to be non-compliant with medication [25]. A French study showed that professionally employed patients had lower adherence of diabetes medication [38] as they were more likely to forget to take medicine because of their busy work schedule. Previous studies from different parts of the world have shown a significant positive association between medication adherence with duration of diabetes [19,23,34,39]. The present study measured medicine intake duration instead of duration of diabetes. However, medicine intake duration had no effect on medication adherence in the study. With regard to types of medicine, an Indian study revealed that adherence was higher with OHA than with insulin alone or with a combination of OHA and insulin [34]. 

Medication adherence was likely to be higher among the Nepalese population presented in the tertiary care hospital in Nepal. The patients who had formal education were more likely to have high medicine adherence compared to those whodid not have formal education. Additionally, patients who attended diabetes counseling had better medication adherence. This implies that a patient with low education should be given special consideration by a concerned individual—either medical personnel or family members—regarding medication of the patient. Similarly, more diabetes counseling services should be facilitated in the country to create awareness among the patients and their family members about diabetes and its overall management including medication intake. Diabetes counseling in a hospital setting or community setting would be beneficial for the patients with diabetes. 

There were both strengths and limitations in the present study. Some of the strengths were as follows: first, diabetes medication adherence was measured using a standardized toolMMAS-8 (© 2006 Donald E. Morisky), after obtaining its license for the Nepali translated version. Second, multiple confounders were controlled using multivariate logistic regression analysis. Third, Open Data Kit software was used for data collection that resulted in complete information of the participants with minimum data entry error. Fourth, the present study was well powered as we were able to enroll the required sample size calculated using scientific methods. The present study was not void of limitations. This was a hospital-based study using convenience sampling. Hence, the study population may not be representative of the general population. Since it was an observational study with no follow-ups, the stability of adherence cannot be determined. In addition, the adherence was self-reported and might have been affected by social desirability bias. This was a cross-sectional study, so temporality cannot be established. It is possible that the patients with high medication were more likely to attend diabetes counseling rather than diabetes counseling causing high medication adherence. 

## 5. Conclusions

We conclude that there is high adherence to diabetes medication among patients with type 2 diabetes attending the hospital setting in Nepal. Diabetes counseling may have a positive impact on medication adherence. Further, low medication adherence was found among less educated people. Diabetes management program designers should pay more attention to reaching less educated patients, for whom nurse-run counseling on diabetes management could be helpful.

## Figures and Tables

**Figure 1 ijerph-20-01537-f001:**
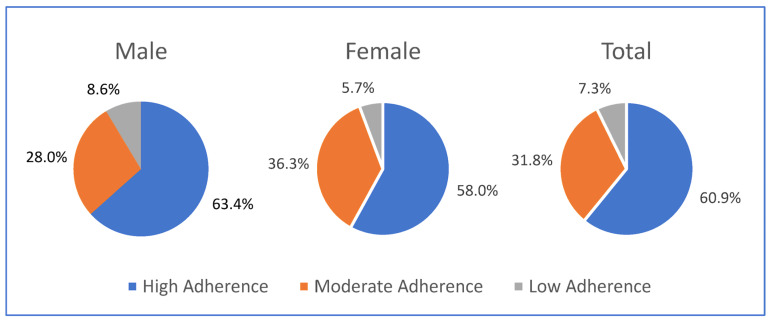
Status of medication adherence. The figure shows the status of medication adherence measured using Morisky medication adherence score (MMAS-8) (© 2006 Donald E. Morisky) (The MMAS (8-item) content, name, and trademarks are protected by US copyright and trademark laws. Permission for use of the scale and its coding is required. A license agreement is available from Donald E. Morisky, ScD, ScM, MSPH, MMAS Research LLC., donald.morisky@moriskyscale.com) and categorized as high adherence (MMAS-8: 8), moderate adherence (MMAS-8: ≥6 or ˂8), and low adherence (MMAS-8: <6). The total number of participants were 343 with 186 male and 157 female. The proportion of high medication was high in all three groups.

**Table 1 ijerph-20-01537-t001:** Sociodemographic characteristics among 343 diabetes patients at Dhulikhel Hospital.

Characteristics	Male (*n* = 186)		Female (*n* = 157)		Total (*n* = 343)	
	*n*	(%)	*n*	(%)	*n*	(%)
**Age, years mean (SD)**	57.1 ± 11.4		54.3 ± 11.4		55.8 ± 11.5	
**Ethnicity**						
Newar	91	(48.9)	77	(49.0)	168	(49.0)
Brahmin/Chhetri	68	(36.6)	60	(38.2)	128	(37.3)
Other	27	(14.5)	20	(12.7)	47	(13.7)
**Marital Status**						
Currently married	179	(96.2)	129	(82.2)	308	(89.8)
Not currently married	7	(3.8)	28	(17.8)	35	(10.2)
**Education**						
No formal education	28	(15.1)	100	(63.7)	128	(37.3)
School Leaving Certificate (SLC)	112	(60.2)	49	(31.2)	161	(47.0)
Above SLC	46	(24.7)	8	(5.1)	54	(15.7)
**Occupation**						
Housewife	0	(0.0)	98	(62.4)	98	(28.6)
Business	64	(34.4)	11	(7.0)	75	(21.9)
Agriculture	44	(23.7)	30	(19.1)	74	(21.6)
Office (professional)	29	(15.6)	5	(3.2)	34	(9.9)
Unemployed	19	(10.2)	5	(3.2)	24	(7.0)
Other occupation	30	(16.1)	8	(5.1)	38	(11.0)
**Residency ***						
Urban	164	(90.6)	132	(88)	296	(89.4)
Rural	17	(9.4)	18	(12.0)	35	(10.6)

* *n* = 331.

**Table 2 ijerph-20-01537-t002:** Clinical Characteristics among 343 diabetes patients at Dhulikhel Hospital.

Characteristics	Male (*n* = 186)		Female (*n* = 157)		Total (*n* = 343)	
	*n*	%	*n*	%	*n*	%
**Types of diabetes medicines**						
OHA	153	(82.3)	136	(86.6)	289	(84.3)
OHA and insulin	22	(11.8)	16	(10.2)	38	(11.0)
Insulin only	11	(5.9)	5	(3.2)	16	(4.7)
**Medicine intake, years** Median (IQR)	3 (1, 6)		2.5 (1, 6)		3 (1, 6)	
**MMAS-8 (© 2006 Donald E. Morisky) Mean (SD) ** ^ **1** ^	7.4 ± 0.9		7.3 ± 1		7.4 ± 1	
**Financial support for diabetes** **medicine ***						
Self	132	(76.3)	27	(18.5)	159	(49.8)
Family member	41	(23.7)	119	(81.5)	160	(50.2)
**Immediate family member with diabetes **						
Yes	29	(15.6)	22	(14)	51	(14.9)
No	157	(84.4)	135	(86)	292	(85.1)
**Attendance diabetes counselling**						
Yes	74	(39.8)	70	(44.6)	144	(42)
No	112	(60.2)	87	(55.4)	199	(58)
**Duration of doctor–patient interaction, minutes mean (SD)**	8.8 ± 5.7		8.3 ± 3.9		8.6 ± 4.9	
Hypertension						
Yes	100	(53.8)	63	(40.1)	163	(47.5)
No	86	(46.2)	94	(59.9)	180	(52.5)
**Self-reported chronic complications**						
Yes	20	(10.8)	9	(5.8)	29	(8.5)
No	165	(89.2)	147	(94.2)	312	(91.5)

* *n* = 319; ^1^ The MMAS (8-item) content, name, and trademarks are protected by US copyright and trademark laws. Permission for use of the scale and its coding is required. A license agreement is available from Donald E. Morisky, ScD, ScM, MSPH, MMAS Research LLC., donald.morisky@moriskyscale.com.

**Table 3 ijerph-20-01537-t003:** Factors associated with medication adherence ^1^ among 343 diabetes patients at Dhulikhel hospital *.

Characteristics	Model 1			Model 2 **			Model 3 ***		
	Bivariate Analysis			Adjusted for Sociodemographic Variables			Adjusted for Sociodemographic & Clinical History		
	(*n* = 343)			(*n* = 343)			(*n* = 343)		
	OR	95% CI	*p*-Value	OR	95% CI	*p*-Value	OR	95% CI	*p*-Value
**Age, years**	0.99	(0.97, 1.01)	0.28	0.99	(0.97, 1.01)	0.40	0.99	(0.97, 1.02)	0.73
**Education**									
No formal education	Ref			Ref			Ref		
Formal education	2.43	(1.55, 3.81)	<0.001	2.36	(1.32, 4.23)	<0.001	2.43	(1.34, 4.39)	0.003
**Occupation**									
Unemployed	Ref			Ref			Ref		
Employed	1.19	(0.76, 1.88)	0.44	0.71	(0.39, 1.29)	0.26	0.70	(0.38, 1.29)	0.25
**Medicine intake duration, Years (nat.log) **	1.00	(0.95, 1.04)	0.85	-	-	-	0.91	(0.73, 1.13)	0.41
**Attendance diabetes counseling**									
No	Ref						Ref		
Yes	1.78	(1.14, 2.79)	0.01	-	-	-	1.76	(1.02, 3.04)	0.04
**Diabetes medicine types**									
Only OHA	Ref						Ref		
Insulin or Insulin with OHA	1.34	(0.73, 2.47)	0.35	-	-	-	1.28	(0.60,2.73)	0.52

* Dependent variable: Medication adherence (high adherence and moderate/low adherence). ** Model 2 adjusted for sociodemographic characteristics (age, education, and occupation);*** Model 3 adjusted for sociodemographic characteristic and clinical characteristics (age, education, occupation, medicine intake duration (natural log), attendance of diabetes counselling and types of diabetes medicines). ^1^ The MMAS (8-item) content, name, and trademarks are protected by US copyright and trademark laws. Permission for use of the scale and its coding is required. A license agreement is available from Donald E. Morisky, ScD, ScM, MSPH, MMAS Research LLC.; donald.morisky@moriskyscale.com.

## Data Availability

Data will be available on reasonable request from the corresponding author.

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
