# Peer review of "Factors Associated with Medication Adherence among Patients with Type 2 Diabetes Mellitus: A Hospital-Based Cross-Sectional Study in Nepal"

_ijerph, 2023, doi:10.3390/ijerph20021537_

Round 1

Reviewer 1 Report

The manuscript entitled " Factors associated with medication adherence among patients  with type 2 diabetes mellitus: A hospital-based cross-sectional study in Nepal." measured medication adherence and its associated factors among type 2 diabetes patients in Nepal with standardized tools, which will help to understand the predictors of medication adherence among type 2 diabetic patients and facilitate the evidence-based management of type 2 diabetes in Nepal. However, some problems must be solved before this review is considered for publication.

1.     In the ABSTRACT part, the authors are suggested to supplement the background of the study, such as the global burden of diabetes and difficulties in diabetes management.

2.     In part 2, Materials and Methods, it would be better if the authors provide criteria for diagnosing type 2 diabetes.

3.     In part 3, RESULTS, the authors should supplement the descriptive analysis of the results of Figure 1. Besides, the supplementary instruction of MMAS (l160-163) and Dependent Variable (L182-186) should be stated in part 2, Materials and Methods.

4.     In part 4, Discussion, it seems paradoxical that the authors discussed the association between older age, diabetes duration and medication adherence after stating that the present study did not show any significant association between them. The authors should read the relevant literature carefully and make the correct discussion.

5.     I recommend that the author add a general summary about the article and its value in the field before the strengths and limitations of the article.

6.     Figure 1 is suggested to be improved. The values for the proportions of each adherence should be located directly above the column. Moreover, compared to the histogram, a pie chart might be more proper to clearly reflect the intuitive proportion of the overall portion.

7.     Some minor mistakes need to be corrected. “at least” rather than “atleast”(L27 and L86); “Strengthening” rather than “STengthening”(L142). In addition, please provide the full name of the abbreviation when it first appears, such as USD (L48) and DM (L169 and L185). Please check the manuscript carefully to avoid the similar mistakes.

Reviewer 2 Report

This is an original study evaluating the factors affecting on medication adherence in Nepal. this study is well-designed, but has minor novelty for worldwide readers (it may be at importance for local readers in Nepal). in my point view, it can be acceptable at lower preference.

minor comments:

why this manuscript has not been published yet? (this study was conducted in 2016)

the low word count of this manuscript suggests that this manuscript is a brief report. 

Reviewer 3 Report

The study by Pushpanjali Shakya et al. ‘’Factors associated with medication adherence among patients with type 2 diabetes mellitus: A hospital-based cross-sectional study in Nepal’’ evaluate factors associated with diabetes medication adherence among the Nepalese population in the hospital setting. The topic of the study is of high importance as diabetes is considered as major public health problem with increasing incidence worldwide. The authors studied and discussed factors that could impact the adherence of diabetes patients to diabetes medication.

There are some concerns:

1.     Under the material and method section, study population and sample size sub-section, you mentioned how sample size was calculated. Could you please insert the formula in the main text or somewhere?

2.     If I am not mistaken, you mentioned that there is one previous study that assessed the diabetes medication adherence in Nepal. If there is why you used proportion of high medication adherence among glycaemic control as 89.60% which is from Israel?  

3.     Could you please elaborate ‘’ratio of non-adherence as 1.15’’ why you used 1.15?

4.     In line 223, older age has been shown to be associated with lower........ how older age was defined? Older age is >50 or >60 or >70? 

Round 2

Reviewer 1 Report

This manuscript can be accepted and published.